# Gender Differences in Factors Associated with the Total Delay in Treatment of Pulmonary Tuberculosis Patients: A Cross-Sectional Study in Selangor, Malaysia

**DOI:** 10.3390/ijerph19106258

**Published:** 2022-05-21

**Authors:** Kee Chee Cheong, Sumarni Mohd Ghazali, Ahmed Syahmi Syafiq Md Zamri, Yoon Ling Cheong, Nuur Hafizah Md. Iderus, Tharmarajah Nagalingam, Qistina Ruslan, Mohd Azahadi Omar, Ahmad Faudzi Yusoff

**Affiliations:** 1Sector for Biostatistics and Data Repository, National Institutes of Health, Ministry of Health Malaysia, Shah Alam 40170, Selangor, Malaysia; drazahadi@moh.gov.my; 2Biomedical Epidemiology Unit, Institute for Medical Research, National Institutes of Health, Ministry of Health Malaysia, Shah Alam 40170, Selangor, Malaysia; sumarni.mg@moh.gov.my (S.M.G.); syahmi.syafiq@moh.gov.my (A.S.S.M.Z.); nuurhafizah@moh.gov.my (N.H.M.I.); qistina@moh.gov.my (Q.R.); 3Biomedical Museum, Institute for Medical Research, National Institutes of Health, Ministry of Health Malaysia, Shah Alam 40170, Selangor, Malaysia; cheongyl@moh.gov.my; 4Kuala Lumpur Hospital, Ministry of Health Malaysia, Shah Alam 40170, Selangor, Malaysia; tharma69@yahoo.com; 5SEAMEO TROPMED Malaysia, Institute for Medical Research, National Institutes of Health, Ministry of Health Malaysia, Shah Alam 40170, Selangor, Malaysia; ahmadfaudzi.y@moh.gov.my

**Keywords:** pulmonary tuberculosis, delay, gender, stigma

## Abstract

*Background*: Gender plays a significant role in health-care-seeking behavior for many diseases. Delays in seeking treatment, diagnosis, and treatment for pulmonary tuberculosis (pTB) may increase the risk of transmission in the community and lead to poorer treatment outcomes and mortality. This study explores the differences in factors associated with the total delay in treatment of male and female pTB patients in Selangor, Malaysia. *Methods*: A cross-sectional study was conducted from January 2017 to December 2017. Newly diagnosed pTB patients (≥18 years) were recruited from selected government health clinics and hospitals in Selangor during the specified study period. An interviewer-administered questionnaire was used to collect information on sociodemographic characteristics, lifestyle, knowledge about pTB, stigma, distance to the nearest health facility, and chronology of pTB symptom onset, diagnosis, and treatment. The total delay was measured as the length of time between the onset of pTB symptoms to treatment initiation. Factors significantly associated with a longer total delay among men and women were identified using binary logistic regression. *Results*: A total of 732 patients (61.5% men, 38.5% women) were enrolled in the study. The median total delay was 60 days. Men who have weight loss as a symptom (AOR: 1.63, 95%CI: 1.10–2.41) and are employed (1.89, 1.15–3.11) were more likely to have a longer total delay, while those who know others who have had pTB (0.64, 0.43–0.96) were less likely to have a longer total delay. On the other hand, among women, having a stigma towards TB (0.52, 0.32–0.84) and obtaining a pTB diagnosis at the first medical consultation (0.48, 0.29–0.79) were associated with a shorter total delay. *Conclusion*: Factors associated with the total delay in pTB treatment were different for male and female pTB patients. Increasing awareness of pTB symptoms and the importance of seeking early medical consultation and a prompt diagnosis among the general public may reduce total delay in pTB treatment.

## 1. Background

While the implementation of the Direct Observation Treatment, Short Course (DOTS) strategy is widespread, the incidence of TB in many countries is continuously rising or remains stagnant, which does not bode well for achieving the global goal of ending TB. The TB incidence rate in Malaysia has increased rather than decreased over the past two decades [1], despite the availability of effective treatment that is provided at no charge for Malaysian citizens, and for a nominal fee for non-citizens, in government healthcare facilities.

Delayed treatment could be one of the factors contributing to the persistence of TB in Malaysia and other countries, and results in increased disease severity and mortality [2]. Delays in seeking treatment, diagnosis, and the initiation of treatment among patients affected with pulmonary tuberculosis (pTB) may also prolong the infectious period and increase the risk of transmission in the community. The total delay period can be divided into the patient delay (defined as the length of time between the onset of symptoms and presentation to a healthcare provider) and health system delay periods (between the date of presentation to healthcare provider and the initiation of tuberculosis treatment). It can also be divided into a diagnostic delay (between the onset of symptoms and labelling of the patient as a tuberculosis patient) and a treatment delay (between tuberculosis diagnosis and the initiation of anti-tuberculosis drugs) [3]. Early detection and effective treatment, the two key factors for successful tuberculosis control, can be achieved by minimising patient delay (the time from the appearance of the first symptom to arriving in standard health care) and health system delay (time between first visit and diagnosis) [4].

A review of past literature show that factors related to treatment delay are age, gender, marital status, employment status, financial constraints, fear of being stigmatised or stigma against TB, knowledge and perception of pTB, logistics, poor social support, presence of certain symptoms, and choice of healthcare provider, as well as the co-existence of a chronic cough and/or other lung diseases, having pulmonary or extrapulmonary tuberculosis, seeking traditional treatments or private practitioners first, and a history of previous TB infection and sputum smear status [5,6,7,8]. However, findings from qualitative and quantitative research indicate that there are gender differences in the factors related to the various types of delay.

Being perceived as a man or a woman may trigger different responses from clinicians, who might then diagnose and suggest interventions differently according to gender [9]. Furthermore, the gender of a patient determines access to healthcare, health-seeking behaviour, and individual use of the health-care system [10,11]. Women affected by TB are more likely to first seek treatment from traditional practitioners or healers [12,13] or low-level, non-hospital facilities such as village clinics and drug stores [14], which lack the necessary diagnostic and medical facilities, subsequently leading to a delayed diagnosis and treatment. Women in Africa and Asia face more barriers (less access to a healthcare provider, financial dependency, stigma, poor health literacy) and, therefore, have longer delays in TB treatment than men [15]. In the South East Asian, Western Pacific, and African regions, men delay due to fears of a loss of income from being away from work while undergoing treatment. On the other hand, among women, a lack of financial autonomy due to being less prioritised for receiving healthcare and stigma (driven by a fear of divorce, reduced marital prospects, or infertility), poor knowledge of TB, and a reliance on traditional healers or general practitioners contribute to the delay [16]. 

In light of this, we conducted this study with the aim to determine whether there are gender differences in the factors associated with the total delay in treatment of pulmonary tuberculosis patients in Selangor, Malaysia, which would inform the planning and provision of TB healthcare services that cater to the different needs of men and women and in designing appropriate gender-specific interventions for reducing delay.

## 2. Methods

### 2.1. Study Design

We conducted a cross-sectional study among newly diagnosed pTB patients from January 2017 to December 2017. Patients were recruited from 75 government primary health clinics and seven hospitals (Ampang Hospital, Sungai Buloh Hospital, Selayang Hospital, Sabak Bernam Hospital, Tanjung Karang Hospital, Kajang Hospital, and Serdang Hospital) in the state of Selangor, which is the most developed state in Malaysia and is located in the central region of Peninsular Malaysia (Appendix A). All of the clinics have the capability to perform sputum smear microscopy for TB and tuberculin skin testing, but only the hospitals and 24 of the health clinics are equipped with X-ray imaging facilities in addition to the aforementioned TB diagnostics. Management of TB cases in all government health facilities are conducted in accordance with clinical practice guidelines drawn by the Ministry of Health Malaysia [17]. 

The sample size for estimating the prevalence of delay was determined using a sample size calculator [18], with an input of a 40% prevalence of delay, an alpha of 0.05, and a 5% precision [19]. Based on figures from the year 2013 for the number of newly diagnosed TB patients in Selangor of 4148 cases (unpublished data), the minimum required sample size was 339. With an additional 20% added for non-response, the final sample size needed was 407. Appendix A shows the distribution of the sample size for each district proportionate to total number of pTB cases reported in 2013 from each district.

### 2.2. Study Participant Recruitment

Newly diagnosed adult pTB cases (age 18 years and above) were identified from the TB patient registries at the study sites and approached to be recruited in the study. Patients undergoing retreatment due to relapse, treatment failure, or defaulted treatment, as well as patients transferred out of the state, those with extrapulmonary TB, and mentally-ill or mentally-challenged pTB patients were excluded. Eligible patients were educated on the risks, benefits, and processes involved in the study and written consent was obtained from consenting patients prior to enrolment and data collection. The protocol for this study was approved by the Medical Research and Ethics Committee of the Ministry of Health of Malaysia (NMRR-16-978-30978).

### 2.3. Data Collection Tools and Procedure

Face-to-face interviews of the patients were conducted with the aid of a pre-tested, structured questionnaire. Information in the questionnaire was organized into several sections: (1) sociodemographic characteristics of the patients, which consisted of age, gender, education level, employment, marital status, residential area, personal income, monthly household income, nationality, and number of households in the same house; (2) Lifestyle behaviours (smoking status and alcohol consumption); (3) Knowledge about TB and stigma; (4) Laboratory and clinical findings; (5) Chronology of symptom onset and treatment. The questionnaire was pre-tested among the data collectors for ease of administration and to ensure that all the information obtained from the interview and review of medical records can be captured in the questionnaire.

The patients’ knowledge on TB was assessed using a seven-item questionnaire: knowledge on TB diagnosis (one item), mode of transmission of TB (two items), treatment prognosis (one item), vaccine for TB (one item), treatment duration (one item), and type of anti-TB drugs (one item). Patients were given a score of one for each correct answer and zero points for incorrect answers or if the patients responded as having no knowledge, and their total score was calculated. The total scores were then classified as ‘good’ (total score ≥ median score) or ‘poor’ TB knowledge (total score < median score). 

For measuring TB stigma, a 14-item scale was used [3]. The scale assessed feelings of shame after pTB diagnosis (one item), reluctance to disclose diagnosis (one item), self-isolation (one item), perceived adverse effects on social, marital, and family relationships (three items), perceived negative health effects such as infertility, complications during pregnancy, pregnancy outcome, and breast feeding (four items), perceived adverse effects on work performance and family responsibilities (two items), reduced chances of marriage (one item), and perceived cost of pTB treatment (one item). Each item is scored on a five-point Likert scale ranging from one (strongly disagree) to five (strongly agree). Patients were classified into two groups (low and high stigma) using the median score as the cut-off for the classification. 

Information on dates of laboratory tests (chest x-ray, sputum smear, sputum culture, and Gen-expert) performed before the diagnosis of pTB, as well as the type of facility where the tests were performed, test results, and date received, in addition to diabetes mellitus and HIV/AIDS status, were retrieved.

Presentation variables were captured consisting of the date of onset of TB-related symptoms, presence of any cardinal symptoms of pTB, and health care seeking behavior, including action after onset of pTB-related symptoms, health facilities patients first went to, type of symptom that made patients seek healthcare, type of healthcare facilities patients sought before commencing TB treatment, when the diagnosis was made and treatment started, and the duration from first symptom onset to first medical consultation. 

The location of the patients’ residential address, hospitals, and government health clinics in Selangor were geocoded using Google Maps and imported into Quantum GIS (QGIS) software. The direct distance from the patients’ home address to the nearest government healthcare facility was determined using the distance matrix tool in QGIS. Eight patients with no house addresses were excluded from the analysis.

### 2.4. Definition of Total Delay 

Total delay in pTB treatment was defined as the sum of patient delay (interval between the onset of TB-related symptoms and the first medical consultation), diagnostic delay (interval between first medical consultation and obtaining a TB diagnosis), and treatment delay (interval between TB diagnosis and the initiation of treatment). The symptoms of pTB include a cough of more than 2 or 3 weeks with/without sputum, chest pain, fever, unexplained weight loss, loss of appetite, and haemoptysis. The median total delay was used as a cut-off to classify patients into delayers (≥median) and non-delayers (<median).

### 2.5. Statistical Methods

A descriptive statistical analysis was conducted to describe the sociodemographic characteristics, lifestyle factors, presence of comorbidities, clinical presentation, knowledge about pTB, stigma, and treatment-seeking behavior of the patients stratified by gender. Differences in the characteristics of male and female patients were analysed using a Pearson’s chi-square test for categorical variables and a Mann–Whitney test for continuous variables. A gender-stratified univariable analysis of associations between sociodemographic characteristics, lifestyle, presence of comorbidities, clinical presentation, knowledge about pulmonary tuberculosis, stigma, and health-seeking behaviour of the patients with the total delay were performed. A simple logistic regression was performed for selected variables identified from the univariable analysis as factors associated with delay, followed by a multiple logistic regression to adjust for confounding. The Hosmer–Lemeshow test was applied to assess the goodness-of-fit of the final multivariable model. The final model was examined for all potential two-way interactions between the identified factors. All statistical analyses were performed using IBM SPSS Statistics for Windows version 26.0 (Armonk, NY, USA: IBM Corp, 2019).

## 3. Results

A total of 732 patients (61.5% men) were enrolled in the study. Almost half of the patients were aged below 40 years old. The majority of the patients were Malays (57.5%), obtained a secondary education (50.8%), were married (62.3%), and were currently employed (72.3%). About 44.4% of the patients had smoked and only 3.7% engaged in drinking. The proportion of patients diagnosed with diabetes and HIV/AIDS was 38.2% and 1.9%, respectively, and 85.2% were sputum smear-positive. The top three self-reported chief complaints of pTB symptoms were a cough of more than 2 weeks (productive and non-productive) (92.3%), loss of weight (62.2%), and fever (58.2%) (Table 1). Figure 1 shows the distribution of patients’ home and public healthcare facilities in the state of Selangor.

The median direct distance from home to the nearest government healthcare facility was approximately 2.1 km. Approximately two-thirds of the patients did not know anyone who had ever been diagnosed with pTB from among their family members, close relatives, work colleagues, friends, or neighbors. After the first onset of pTB symptoms, most of the patients (81.1%) sought treatment from a healthcare practitioner, followed by self-medication (10.5%) and possessed medicine from a pharmacy without prescriptions (8.2%). The type of healthcare facility sought after the onset of symptoms were general practitioner (37.2%), government health clinic (31.6%), government hospital (26.6%), private hospital (4.2%), and traditional and complementary medicine practitioner (0.3%). Almost half, 43.3%, of the patients received their pTB diagnosis during their first medical consultation (Table 1).

There were differences in sociodemographic characteristics (age, ethnicity, marital status, and employment status) and lifestyle (alcohol consumption and smoking) factors between men and women. There were no significant differences in self-reported symptoms of pTB between men and women, except for loss of weight (*p* < 0.001) and loss of appetite (*p* = 0.035). In addition, there were no significant gender differences in knowledge, stigma, and health seeking behavior. The proportion of men (66.2%) diagnosed with having pTB during the first medical consultation was significantly higher compared to women (33.8%) (*p* = 0.020). Overall, the median total delay was 60 days. The patient delay contributed the most to the total delay (median = 30 days), followed by diagnostic delay (median = 6). There was almost no treatment delay (Table 1). The vast majority of the patients (71.4%) were immediately started on TB treatment on the day of diagnosis. Women had a significantly longer total delay (median = 62 days) than men (median = 53 days) (*p* = 0.035), as well as a longer diagnostic delay (median = 7 days vs. median = 4.5 days, *p* = 0.014).

Table 2 shows the results of the univariable analysis of association between sociodemographic, lifestyle factors, clinical features, knowledge about pTB, stigma, and healthcare-seeking behavior with delay, stratified by gender. In men, employment status (*p* = 0.010) and having the symptom of weight loss (*p* = 0.012) were significantly associated with the total delay, whereas among women, pTB stigma (*p* = 0.012) and being diagnosed with pTB at the first medical consultation (*p* = 0.003) were significantly associated with the total delay.

A multivariable analysis showed that, among men, having weight loss as a chief symptom (AOR: 1.63, 95%CI: 1.10–2.41) and being employed (1.89, 1.15–3.11) were associated with higher odds, and knowing someone with pTB (0.64, 0.43–0.96) was associated with lower odds of a total delay > 60 days among men. On the other hand, in women, those with pTB stigma (0.52, 0.32–0.84) and obtaining a pTB diagnosis at the first medical consultation (0.48, 0.29–0.79) were associated with less delay (Table 3).

## 4. Discussion

Our study showed that, in men, having weight loss as the chief symptom and being employed were significantly associated with a longer total delay, and knowing someone with pTB was significantly associated with a shorter total delay. On the other hand, in women, pTB diagnosed at the first medical consultation and having pTB stigma were significantly associated with a shorter total delay.

The median total delay in the present study (60 days, IQR: 66) was shorter than two older studies that investigated pTB patients in two TB referral hospitals in Malaysia, by Hooi [20] in 1994 (90 days) and Liam and Tang [21] in 1997 (88 days). The shorter delay in our study may be attributed to improvements in TB diagnostic tests, increased accessibility to healthcare facilities, and increased alertness among health care workers and awareness in the community over the past two decades. Among other previous studies using the same definition, the median total delay in a Northwest Ethiopian study was comparable [22], but other studies reported shorter delays, i.e an Indian study found the median was 55.3 days (IQR: 46.5–61.5) [23], Iran (49 days) [24], China (47 days, IQR: 30–62). However, our median total delay is still lower than that reported in England (88 days, IQR: 49–163) [25], Indonesia (65 days, IQR: 37–119) [26], and Colombia [27]. Differences in the length of the total delay between these studies are probably due to the heterogeneity in study design and sample size, as well as the health care setting, pTB incidence rate, sociodemographic, cultural, and economic status of the study populations, and the time period in which the studies were conducted.

Our study revealed that women had a significantly longer total delay compared to men. Findings of previous research on gender associations in relation to total delay have been inconsistent. A multi-country study conducted by the WHO office for the eastern Mediterranean region in seven countries (Egypt, Iran, Iraq, Pakistan, Somalia, Syria, and Yemen) in 2003–2004 showed that there were differences between countries. The mean total delay among Egyptian women was >44 days and was significantly lower compared to Egyptian men (AOR: 0.62, 95%CI: 0.39–0.99), whereas Yemeni women had a mean of >35 days, which was significantly higher than their male counterparts (AOR: 2.29, 95%CI: 1.26–4.14). In the other five countries, no gender differences in the total delay were found [3]. Our finding is consistent with many studies conducted in different settings in China [28], England [25], the Middle East, and North Africa [5]. Some plausible explanations include women usually having a lower socioeconomic status (less educated, fewer employment opportunities, and less income) compared to men, but bear the burden of taking care of other family members and domestic duties on top of their waged work, and thus having less material resources, decision-making autonomy, and time than men to seek early treatment after the onset of pTB-related symptoms [10,16,29]. Furthermore, women are less likely to present with severe symptoms than men, which may be due to biological differences in TB vulnerability and, consequently, may result in a delayed diagnosis [30] (UNDP 2015).

Being employed was shown to be positively associated with a delay among the men in our study, but not among the women. This is similar to a finding from a systematic review of studies in the Middle East and North African countries that concluded that patients who were unemployed were more likely to have a shorter patient delay (AOR: 0.83; 95% CI: 0.72, 0.95) [5]. This could be because men are usually the sole breadwinner or the main contributor to the household income. As such, among employed men, fears of job and income loss may deter them from seeking early medical consultation and treatment [16,29]. Future studies need to explore the gender differences in the relationship between employment status and treatment delay.

In our study, cough, weight loss, and fever were the three most common TB-related symptoms reported. However, only weight loss was significantly associated with a longer delay among men. A study conducted in Italy reported that haemoptysis and weight loss were the two cardinal signs significantly associated with total delay (>45 days) [31]. Another study in Ethiopia reported a significant association between loss of weight (AOR: 2.53, 95% CI: 1.35,4.74) and fatigue (AOR: 2.38, 95% CI: 1.36,4.17) with patient delay (delay ≥30 days) in seeking a health care provider after the onset of symptoms [32]. In a univariable analysis of the effect of gender on clinical characteristics and treatment outcomes among 4867 TB patients in Victoria, Australia (54.5% males) who were diagnosed from 2002–2015 also showed that the odds of weight loss was significantly higher in males than females and exhibited a more severe disease at presentation [33]. One of the possible explanations is that noticeable weight loss was probably due to disease progression and worsening of symptoms after prolonged delay. Male patients may perceive initial weight loss as a non-specific and transient symptom since weight loss could be associated with other illnesses, until the deterioration and manifestation of other specific symptoms such as chest pain and haemoptysis [29]. This is in line with studies by Tedla et al. [34] and Virenfeldt et al. [35], which reported close associations between a long delay (>60 days and 12.1 weeks, respectively) and pTB clinical severity. However, these findings are contradictory to studies in Brazil [36,37], in which patients who delayed seeking treatment after the first onset of symptoms were less likely to be among those who had reported a loss of weight. The authors suggested that the patients’ knowledge about weight loss as one of the symptoms of TB may be a possible explanation for this. Moreover, self-reported profound weight loss may have triggered a physician’s suspicion of TB and initiated early TB-related laboratory investigations [38]. Nevertheless, we could not explain why weight loss was not associated with a delay among women. One hypothesis is weight loss may be less noticeable in obese individuals. The national prevalence of obesity was significantly higher among women 24.7%(95% CI: 22.9–26.6) vs. only 15.3% (95% CI: 13.6–17.0) among men in 2019 [39]; it may be that, within our sample, the female patients are likewise more obese than the males. Further research is needed to investigate the gender differences in the relationship between self-reported TB-related symptoms and a prolonged delay.

Our data showed that male patients who previously knew other people who had pTB had a shorter total delay. Knowing someone who had been infected with TB may raise ones awareness of pTB, which made them more alert and prompted them to seek treatment earlier. A study conducted among 173 tuberculosis patients in Amhara state, Ethiopia reported that those who had never heard about TB had a three-fold increased risk of delayed healthcare seeking (>21 days) [40].

It is generally assumed that stigma towards TB is one of the barriers for prompt diagnosis and treatment of pTB in the community. However, our findings are to the contrary, with women who had tuberculosis-related stigma being more likely to seek early medical care after the onset of symptoms. This is consistent with a previous study by Pungrassami et al. [41], which showed that higher TB stigma was associated with a shorter time from the first TB symptom to the first visit to a qualified healthcare provider among women in southern Thailand, with the explanation that women may be more concerned about the consequences of the illness on their social interaction and are hence more likely to seek early treatment for quick relief of TB symptoms. This explanation is supported by a review of previous studies conducted from 1976 to 2009 that suggest pTB stigma has a different socioeconomic impact on men and women, which eventually affects their willingness to undertake TB screening or to seek early medical consultation after the onset of pTB symptoms [42]. Women were more likely to worry about the negative impact of TB stigma on their marriage prospects, their family roles, and their relationships with family members [43], whereas men tend to be more concerned about employment opportunities and income.

Among women, receiving the TB diagnosis at the first medical consultation was inversely associated with delay. Prompt diagnosis following the first medical consultation with a health care provider significantly shortens total delay. On the other hand, patients who had consulted multiple health care providers for treatment after the onset of symptoms were more likely to incur a health system delay [26,44]. However, the patient may delay the first medical consultation after the onset of symptoms. Our results show that 20.6% (58/282) of female patients sought alternative treatment (self-medication, over the counter medicine, traditional healers) before the first medical consultation, compared to 17.8% (80/450) of male patients. Women are more likely to have a longer delay than men [6] perhaps because they are generally of lower socioeconomic status, are less mobile, and have less decision-making power in allocating family resources [10]. Consequently, they tend to present with more severe illness and more apparent TB clinical manifestations, and are thus more likely to be identified and diagnosed at the first medical consultation.

## 5. Limitations

Our study has several limitations. Self-reported dates of the onset of symptoms collected retrospectively could be inaccurate (recall bias); hence, the duration of delays may be under- or overestimated. Secondly, the questionnaire we used to evaluate the TB stigma and knowledge about pTB was adopted from the WHO Eastern Mediterranean Region study [3], the reliability and validity of which has not been tested in the Malaysian population. This may affect the quality of the data collected, considering the sociocultural and health literacy differences between our study population and the Eastern Mediterranean population for which the questionnaire was originally developed. Further studies should be carried out to develop valid and reliable questionnaires for assessing TB knowledge and stigma for the Malaysian population.

## 6. Conclusions

Our data showed that factors associated with the total delay in pTB treatment were different for male and female pTB patients. Hence, integrating gender-specific interventions in TB prevention and control programs is essential in combating and eliminating TB treatment delay. These programs need to include promoting awareness of pTB symptoms and placing an emphasis on the importance of seeking medical attention early.

## Figures and Tables

**Figure 1 ijerph-19-06258-f001:**
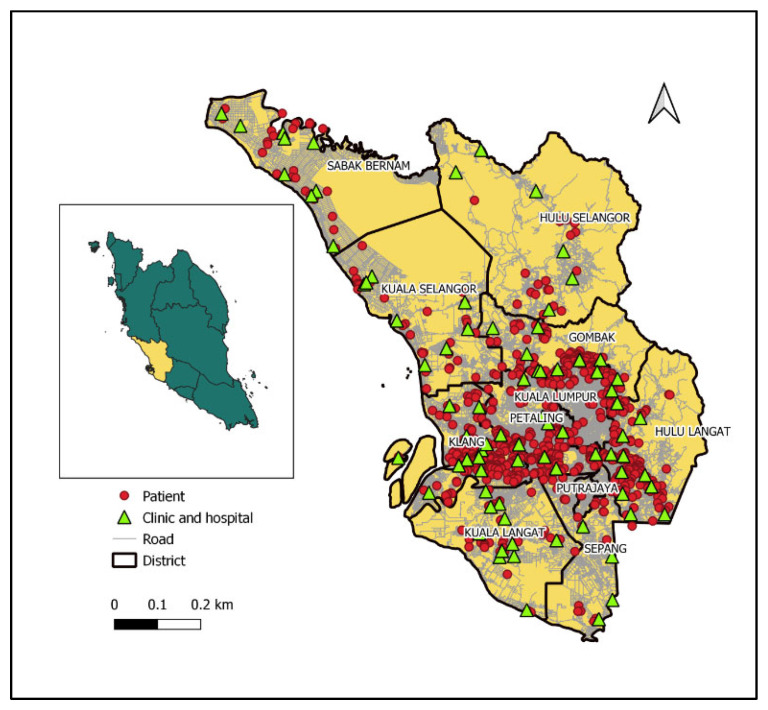
Distribution of pulmonary tuberculosis patients and public healthcare facilities in the state of Selangor, Malaysia.

**Table 1 ijerph-19-06258-t001:** Sociodemographic characteristics, lifestyle, co-morbidities, clinical presentation, knowledge about pulmonary tuberculosis, stigma, and healthcare-seeking behavior of the patients stratified by gender (n = 732).

Variables	Total n (%)	Male (n = 450) n (%)	Female (n = 282) n (%)	*p*-Value
**Sociodemographic characteristics**				
Citizenship				
Malaysian	632 (86.3)	391 (61.9)	241 (38.1)	0.329
Non-Malaysian	100 (13.7)	59 (59.0)	41 (41.0)	
Age (year)				
18–29	210 (28.7)	111 (52.9)	99 (47.1)	0.020
30–39	142 (19.4)	91 (64.1)	51 (35.9)	
40–49	128 (17.5)	78 (60.9)	50 (39.1)	
50–59	146 (19.9)	95 (65.1)	51 (34.9)	
≥60	106 (14.5)	75 (70.8)	31 (29.2)	
Ethnicity				
Malays	421 (57.5)	240 (57.0)	181 (43.0)	0.001
Chinese	78 (10.7)	64 (82.1)	14 (17.9)	
Indians	88 (12.0)	57 (64.8)	31 (35.2)	
Other Indigenous group (Sabah/Sarawak/Orang Asli)	45 (6.1)	30 (66.7)	15 (33.3)	
Others	100 (13.7)	59 (59.0)	41 (41.0)	
Education Level				
No formal education	34 (4.6)	17 (50.0)	17 (50.0)	0.164
Primary	153 (20.9)	103 (67.3)	50 (32.7)	
Secondary	372 (50.8)	230 (61.8)	142 (38.2)	
Tertiary	174 (23.6)	100 (57.8)	73 (42.2)	
Marital status				
Single	233 (31.8)	154 (66.1)	79 (33.9)	0.001
Married	455 (62.3)	275 (60.4)	180 (39.6)	
Divorced	27 (3.7)	18 (66.7)	9 (33.3)	
widowed	17 (2.3)	3 (17.6)	14 (82.4)	
Household size				
≤3	272 (37.2)	181 (66.5)	91 (33.5)	0.095
4–5	266 (36.6)	156 (58.6)	110 (41.4)	
>5	194 (26.5)	113 (58.2)	81 (41.8)	
Employment status				
Unemployed	203 (27.7)	87 (42.9)	116 (57.1)	<0.001
Employed	529 (72.3)	363 (68.6)	166 (31.4)	
**Lifestyle factors**				
Smoking status				
Never smoked	407 (55.6)	140 (34.4)	267 (65.6)	<0.001
Current smoker	190 (26.0)	181 (95.3)	9 (4.7)	
Former smoker	135 (18.4)	129 (95.6)	6 (4.4)	
Alcohol consumption				
No	705 (96.3)	427 (60.6)	278 (39.4)	0.010
Yes	27 (3.7)	23 (85.2)	4 (14.8)	
**Presence of co-morbidities**				
Diabetes	207 (38.2)	135 (65.2)	72 (34.8)	0.191
HIV/AIDS	14 (1.9)	13 (92.9)	1 (7.1)	
pTB Stigma scored, median (IQR)	38 (10)	38 (10)	38 (10)	0.341
pTB knowledge scored, median (IQR)	5 (2)	5 (2)	5 (2)	0.830
**Clinical presentation**				
Sputum smear status				
Positive	624 (85.2)	385 (61.7)	239 (38.3)	0.608
Negative	108 (14.8)	65 (60.2)	43 (39.8)	
Chief Symptom(s) ^a^				
Cough	676 (92.3)	413 (61.1)	263 (38.9)	0.462
Fever	426 (58.2)	253 (59.4)	173 (40.6)	0.171
Night sweat	284 (38.8)	173 (60.9)	111 (39.1)	0.822
Loss of weight	455 (62.2)	266 (58.5)	189 (41.5)	0.035
Loss of appetite	406 (55.5)	225 (55.4)	181 (44.6)	*p* < 0.001
Haemoptysis	167 (22.8)	111 (66.5)	56 (33.5)	0.127
Pleuritic chest pain	166 (22.7)	93 (56.0)	73 (44.0)	0.104
**Accessibility to TB diagnosis and treatment and health seeking behavior**				
Distance measured from home to the nearest public health facilities (meters), median (IQR)	2117.3 (2078.2)	2192.9 (2011.0)	2044.4 (2180.1)	0.565
Know anyone who had PTB				
No	461 (63.0)	292 (63.3)	169 (36.7)	0.176
Yes	271 (37.0)	158 (58.3)	113 (41.7)	
Action after onset of symptoms				
Seek treatment from healthcare practitioner	594 (81.1)	370 (62.3)	224 (37.7)	NA
Self-medicate (take herbs, supplements, OTC drugs)	77 (10.5)	48 (62.3)	29 (37.7)	
Traditional medicine practitioner	1 (0.1)	0 (0)	1 (100.0)	
Seek medication from pharmacy	60 (8.2)	32 (53.3)	28 (46.7)	
Type of healthcare facility first sought after the onset of symptoms				
General practitioner	272 (37.2)	164 (60.3)	108 (39.7)	0.523
Private hospital	31 (4.2)	19 (59.4)	13 (40.6)	
Government health clinic	231 (31.6)	136 (58.9)	95 (41.1)	
Government hospital	195 (26.6)	130 (66.7)	65 (33.3)	
Traditional and complementary medicine practitioner	2 (0.3)	1 (50.0)	1 (50.0)	
PTB was diagnosed the first consultation				
No	415 (56.7)	240 (57.8)	175 (42.2)	0.020
Yes	317 (43.3)	210 (66.2)	107 (33.8)	
Total delay (day), median (IQR)	59 (66)	53 (69)	62 (62)	0.035
Patient delay (day), median (IQR)	30 (50)	30 (50)	30 (53)	0.459
Diagnostic delay (day), median (IQR)	6 (27)	4.5 (20)	7 (31)	0.014
Treatment delay (day), median (IQR)	0 (1)	0 (1)	0 (1)	0.363

Pearson’s chi-square test was performed for categorical variables and the Mann–Whitney test was performed for continuous variables. ^a^ Respondents may have had more than one symptom. NA = not applicable.

**Table 2 ijerph-19-06258-t002:** Univariable analysis of factors with the total delay, stratified by gender.

Variables	Men (n = 450)			Women (n = 282)		
	Total Delay (<60 Days), n (%)	Total Delay (≥60 Days), n (%)	*p*-Value ^b^	Total Delay (<60 Days), n (%)	Total Delay (≥60 Days), n (%)	*p*-Value ^b^
**Overall**	239 (53.1)	211 (46.9)	-	132 (46.8)	150 (53.2)	-
**Sociodemographic characteristics**						
Citizenship						
Malaysian	208 (53.2)	183 (46.8)	0.925	111 (46.1)	130 (53.9)	0.540
Non-Malaysian	31 (52.5)	28 (47.5)		21 (51.2)	20 (48.8)	
Age (year)						
18–29	58 (52.3)	53 (47.7)	0.735	47 (47.5)	52 (52.5)	0.829
30–39	49 (53.9)	42 (46.2)		21 (41.2)	30 (58.8)	
40–49	37 (47.4)	41 (52.6)		23 (46.0)	27 (54.0)	
50–59	51 (53.7)	44 (46.3)		27 (52.9)	24 (47.1)	
≥60	44 (58.7)	31 (41.3)		14 (45.2)	17 (54.8)	
Ethnicity						
Malays	123 (51.2)	117 (48.8)	0.587	85 (47.0)	96 (53.0)	0.372
Chinese	32 (50.0)	32 (50.0)		5 (35.7)	9 (64.3)	
Indians	34 (59.6)	23 (40.4)		17 (54.8)	14 (45.2)	
Other Indigenous group (Sabah/Sarawak/Orang Asli)	19 (63.3)	11 (36.7)		4 (26.7)	11 (73.3)	
Others	31 (52.5)	28 (47.5)		21 (51.2)	20 (48.8)	
Education level						
No formal education	7 (41.2)	10 (58.8)	0.528	6 (35.3)	11 (64.7)	0.483
Primary	52 (50.5)	51 (49.5)		20 (40.0)	30 (60.0)	
Secondary	122 (53.0)	108 (47.0)		71 (50.0)	71 (50.0)	
Tertiary	58 (58.0)	42 (42.0)		35 (47.9)	38 (52.1)	
Marital status						
Single	73 (47.4)	81 (52.6)	0.08	32 (40.5)	47 (59.5)	0.186
Married/divorced/widowed	166 (56.1)	130 (43.9)		100 (49.3)	103 (50.7)	
Household size						
≤3	101 (55.8)	80 (44.2)	0.144	43 (47.3)	48 (52.7)	0.456
4–5	87 (55.8)	69 (44.2)		47 (42.7)	63 (57.3)	
>5	51 (45.1)	62 (54.9)		42 (51.9)	39 (48.1)	
Employment status						
Unemployed	57 (65.5)	30 (34.5)	0.010	55 (47.4)	61 (52.6)	0.865
Employed	182 (50.1)	181 (49.9)		77 (46.4)	89 (53.6)	
**Lifestyle factors**						
Smoking status						
Never smoked	83 (59.3)	57 (40.7)	0.106	124 (46.4)	143 (53.6)	NA
Current smoker	86 (47.5)	95 (52.5)		4 (44.4)	5 (55.6)	
Former smoker	70 (54.3)	59 (45.7)		4 (66.7)	2 (33.3)	
Alcohol consumption						
No	231 (54.1)	196 (45.9)	0.071	132 (47.5)	146 (52.5)	NA
Yes	8 (34.8)	15 (65.2)		0 (0)	4 (100.0)	
**Presence of co-morbidities**						
Diabetes	69 (51.1)	66 (48.9)	0.578	36 (50.0)	36 (50.0)	NA
HIV/AIDS	6 (46.2)	7 (53.8)	0.610	1 (100.0)	0 (0)	
**Clinical presentation**						
Sputum smear status						
Positive	207 (53.8)	178 (46.2)	0.498	116 (48.5)	123 (51.5)	0.171
Negative	32 (49.2)	33 (50.8)		16 (37.2)	27 (62.8)	
Presence of chief symptom (s) ^a^						
Cough	219 (53.0)	194 (47.0)	0.905	122 (46.4)	141 (53.6)	0.598
Fever	138 (54.5)	115 (45.5)	0.490	84 (48.6)	89 (51.4)	0.459
Night sweat	90 (52.0)	83 (48.0)	0.741	50 (45.0)	61 (55.0)	0.633
Loss of weight	128 (48.2)	138 (51.9)	0.012	88 (46.6)	101 (53.4)	0.905
Loss of appetite	111 (49.3)	114 (50.7)	0.118	79 (43.6)	102 (56.4)	0.154
Haemoptysis	65 (58.6)	46 (41.4)	0.177	22 (39.3)	34 (60.7)	0.208
Pleuritic chest pain	45 (48.4)	48 (51.6)	0.316	39 (53.4)	34 (46.6)	0.188
**Knowledge of pTB**						
Poor (≥median score)	130 (51.4)	123 (48.6)	0.407	72 (46.2)	84 (53.8)	0.933
Good (<median score)	103 (55.4)	83 (44.6)		56 (46.7)	64 (53.3)	
**pTB stigma**						
No (<median score)	135 (57.4)	100 (42.6)	0.054	56 (39.4)	86 (60.6)	0.012
Yes (≥ median score)	104 (48.4)	111 (51.6)		76 (54.3)	64 (45.7)	
**Accessibility to TB diagnosis and treatment and health seeking behavior**						
Distance to the nearest Government health facilities						
Distance < median	109 (50.2)	108 (49.8)	0.237	71 (47.7)	78 (52.3)	0.764
Distance ≥ median	130 (55.8)	103 (44.2)		61 (45.9)	72 (54.1)	
Know anyone who had PTB						
No	146 (50.0)	146 (50.0)	0.072	84 (49.7)	85 (50.3)	0.233
Yes	93 (58.9)	65 (41.1)		48 (42.5)	65 (57.5)	
Action after onset of symptoms						
Seek treatment from healthcare practitioner	197 (53.2)	173 (46.8)	0.437	108 (48.2)	116 (51.8)	NA
Self-medicate (take herbs, supplements, OTC drugs)	28 (58.3)	20 (41.7)		10 (34.5)	19 (65.5)	
Traditional medicine practitioner	-	-		1 (100.0)	0 (0)	
Seek medication from pharmacy	14 (43.8)	18 (56.2)		13 (46.4)	15 (53.6)	
Type of healthcare facility first sought after the onset of symptoms						
General practitioner	83 (50.6)	81 (49.4)	NA	45 (41.7)	63 (58.3)	NA
Private hospital	12 (63.2)	7 (36.8)		7 (53.8)	6 (46.2)	
Government health clinics	65 (47.8)	71 (52.2)		49 (51.6)	46 (48.4)	
Government hospital	79 (60.8)	51 (39.2)		30 (46.2)	35 (53.8)	
Traditional and complementary medicine practitioner	0 (0)	1 (100.0)		1 (100.0)	0 (0.0)	
PTB was diagnosed at the first consultation						
No	118 (49.2)	122 (50.8)	0.073	70 (40.0)	105 (60.0)	0.003
Yes	121 (57.6)	89 (42.4)		62 (57.9)	45 (42.1)	

^a^ Respondents may have had more than one symptom. ^b^ Pearson’s Chi-square test. NA = not applicable.

**Table 3 ijerph-19-06258-t003:** Factors associated with total delays in pulmonary tuberculosis treatment by gender.

Variables	Crude OR (95% CI)	*p* Value	Adjusted OR (95% CI)	*p* Value
**All** ^a^				
Marital status				
Single	1		1	
Married/divorced/widowed	0.72 (0.53, 0.98)	0.038	0.70 (0.51, 0.96)	0.027
Alcohol consumption (Consumed alcohol in the past 7 days)				
No	1		1	
Yes	2.52 (1.09, 5.8)	0.031	2.59 (1.10, 6.07)	0.029
Loss of appetite				
No	1		1	
Yes	1.41 (1.05, 1.89)	0.021	1.42 (1.05, 1.91)	0.021
PTB was diagnosed the first consultation				
No	1		1	
Yes	0.61 (0.45, 0.82)	0.001	0.61 (0.46, 0.83)	0.001
**Men** ^b^				
Employment status				
No	1		1	
Yes	1.85 (1.13, 3.03)	0.014	1.89 (1.15, 3.11)	0.013
Loss of weight as chief symptoms				
No	1		1	
Yes	1.75 (1.18, 2.61)	0.006	1.63 (1.10, 2.41)	0.015
Know anyone who had pTB				
No	1		1	
Yes	0.69 (0.46, 1.03)	0.067	0.64 (0.43, 0.96)	0.031
**Women** ^c^				
PTB diagnosed on the first consultation				
No	1		1	
Yes	0.49 (0.29, 0.82)	0.007	0.48 (0.29, 0.79)	0.004
pTB stigma				
No	1		1	
Yes	0.57 (0.35, 0.92)	0.021	0.52 (0.32-0.84)	0.008

^a^ Multiple logistic regression analysis (Forward Stepwise [likelihood ratio]) was performed adjusted for other variables in the model. Classification table = 57.6%, Hosmer-Lemeshow test for goodness of fit showed *p* = 0.299. ^b^ Multiple logistic regression analysis (Forward Stepwise [likelihood ratio])was performed adjusted for other variables in the model. Classification table = 59.63%, Hosmer-Lemeshow test for goodness of fit showed *p* = 0.285. ^c^ Multiple logistic regression analysis (Forward Stepwise [likelihood ratio])was performed adjusted for other variables in the model. Classification table = 61.2%, Hosmer-Lemeshow test for goodness of fit showed *p* = 0.254.

## Data Availability

The datasets used and analysed during the current study are available from the corresponding author upon reasonable request.

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
