# Peer review of "Gender Differences in Factors Associated with the Total Delay in Treatment of Pulmonary Tuberculosis Patients: A Cross-Sectional Study in Selangor, Malaysia"

_ijerph, 2022, doi:10.3390/ijerph19106258_

Round 1

Reviewer 1 Report

Thanks for recommending me as a reviewer. In this paper, authors explores the differences in factors associated with total delay in treatment of male and female pTB patients in Selangor, Malaysia. As authors complete minor revisions, the quality of the study will be further improved.

  1. The introduction section is well written. If the author describes in more detail why it is important to analyze gender differences in the introduction, it can help readers understand.

2. line 89: The authors specifically described sampling in the Methods section.

3. line 178: "All statistical analyses were performed using SPSS for Window." - Please refer to the form journal and add the source for the tool. - 

4. line 336-: It is recommended to separate the "limitations" section separately.

Reviewer 2 Report

This study addresses causes and associated factors of total diagnostic and treatment delay in pulmonary tuberculosis patients, in a cross-sectional design, conducted in 82 different TB treatment centers in a specific area (Selangor, with >6M population) within peninsular Malaysia. The methods were derived from a multi-country study conducted in de Middle East in 2004-5. The data were all sampled using questionnaires, obtained from study participants recruited among patients recently diagnosed with pulmonary tuberculosis. No objective measurements like body mass index, imaging studies, with analysis of disease severity were used in their multivariate model to predict total time lapsed between first symptoms and initiation of TB treatment.

In general, it is highly useful to repeat a study exploring risk factors for total delay in a different setting, and a different time period; therefore, the study data are potentially highly relevant for national program managers and treatment centers around the world, to try reduce delay. The authors correctly state in their introduction that delay is bad for patients themselves, as delay may be harmful, in causing more sequelae with even inherent risk of mortality as well as morbidity; and delay is also potentially harmful at the community level, as TB transmission can only stop if the diagnosis is made and effective treatment is started.

Major points:

1) The authors' summary of the different components of delay is slightly confusing; the total delay is diagnostic plus therapeutic; this total time can also be divided in patient delay (i.e., the time between first onset of symptoms and first contact with the health care system) and doctors' or health care delay, which is in part, diagnostic delay; and delay between diagnosis and first day of treatment; this is therapeutic delay; mentioning four different forms of delay is somewhat confusing, as these two devisions are in parallel - there are not four different consecutive episodes in delay . . The graphic shown in the paper of ref 3 clarifies this point very well. Please consider rephrasing lines 51-56.

2) The authors describe  the different aspects of delay but fail to tease these components apart in their analysis.  This is a pity, as shortening patient delay should be achieved in a very different way than health system delay: the former would require health education to the public; or fighting stigma; while the latter would require training of health care personnel, and investing in health care improvements (diagnostic imaging and microbiological lab facilities; training and continuous medical education of doctors and nurses; improving pharmacy logistics; etc).  Combining all components of delay results in a lack of focus for potential improvements. Stating that women that had their TB diagnosis in their first medical consultation resulted in reduced delay (line 327) is not very meaningful, as this is highly self-evident: if the diagnosis is reached early, we will obviously have little delay . . 

3) several different potential study participants were excluded; especially, those with mental illness and mental challenges (line 112). Please provide a flow chart to allow the reader to understand how many patients were screened; and how many were excluded, for which specific reasons, so as to allow readers to appreciate the selection in recruitment.

4) the sample size required was calculated as 407; while the total number of participants was 632; please explain why the final sample size was 155% of the calculated sample. 

5) self-reported wight loss in women was not associated in delay, in their multivariable model. Are women in the study area slim, or generally obese? Self reported wight loss might be unreliable; if obese women loose weight, this might even be considered beneficial among obese women, while this would be different if BMI would be reduced to <18.5; please try to bring information to readers unfamiliar with the setting of peninsular Malaysia.

Minor points;

6) line 49; 'infected with pTB' . . affected with pTB; infection in TB is usually asymptomatic, e.g., skin test or IGRA conversion. Avoid 'infection' to avoid confusion . .

7) line 51: rephrase 'four types of'.  See above, under 1)

8) line 67: 'While'. erase "while' - incorrect English style. If erased, the sentence is okay.

9) line 124:  . . knowledge on TB stigma and presentation: knowledge about TB; stigma; and presentation of symptoms.

10) 5-point Likert scale: 0-1-2-3-4-5 would be a 6-point scale! please rephrase, or explain . . . . In the original paper form 2006, a 5-point scale was used: 0-1-2-3-4

11) line 145:  . .cvaptured, consisting of (not consisted)

12) line 146: treatment-seeking behavior: try to use one single expression; I suggest to use 'health care seeking behavior' throughout .. 

13) line 173 Simple logistic regression and were performed:  . . regression was performed

14) line 178: SSPS - which version? (mention the town, and the year of issue)

15)line 197 (also in table 1, bottom, page 6): TCM practitioner is probably a traditional healer; avoid abbreviation, and spell out

16) table 1 top of page 6: distance from home: this distance is listed twice; delete in the top; add the metrics of the distance (?reported or measured???) is it in m? 

17) line 209: there no: that were no (add the verb)

18) line 212: women had significantly more (not significant)

19) line 218 dan diagnosed: delete 'dan'

20) line 233: in men, self-reported of weight loss: delete 'of'

21) line 236 delete 'Whereas'

22) line 236 women those were diagnosed: women that were diagnosed .. 

23) line 260: and which . . delete with, use comma instead

24) line 264  . .explanations are women . .. .explanations are, that women . .

25) line 271  ..  shown be.. shown to be .. 

26) line 299: to be among those who had . . (add who)

27) line 303: this seems self-evident; is doctors reach the pTB diagnosis early, delay will be short . . see comment 2

Reviewer 3 Report

Dear authors, congratulations for this important research you've made. pTB is a phenomena that still requires more knowledge, although it is an old issue in public health. 

Your manuscript is well structured and the methodological process is well explained, as well as the results presented. 

I just have two suggestions:

1) in line 118 you mention that one part of your data collection was made with interviews that were conducted with the aid of a pre-tested, structured questionnaire. I believe it would be better if you explain the organization of the questionnaire, and also how the pre-test was conducted. 

2) In your results, line 341 you mention that "the reliability and validity of [the questionaire you used]  has not been tested in the Malaysian population". Can't your study contribute to the adaptation to Malaysian population? Don't you want to mention it in the conclusions?

Again, congratulations. Just this litle step to make the manuscript more close to perfection! Good work. 

Reviewer 4 Report

Title: Please include in the title the kind of study you have done (cross-sectional study).

Introduction: The introduction is poor in content, it would be important to better frame the issue of gender playing a significant role in treatment-seeking behavior for many diseases with epidemiological data regarding the general context in which this issue is studied and then focus on the epidemiological data

Methods: Well done, description is accurate and complete

Results/ Conclusions: Also in this section of the paper the description of the results is accurate and complete. The paragraph from line 232 to 237 is redundant (“Our study indicates…. with less delay”). I suggest to rewrite it in a more concise way.

Table: clear and intuitive
